# Sopa: a technology-invariant pipeline for analyses of image-based spatial omics

Quentin Blampey [1,2,4] ✉, Kevin Mulder [2,4], Margaux Gardet[2], Stergios Christodoulidis [1], Charles-Antoine Dutertre[2], Fabrice André [2,3], Florent Ginhoux[2] & Paul-Henry Cournède [1] ✉

Spatial omics data allow in-depth analysis of tissue architectures, opening new opportunities for biological discovery. In particular, imaging techniques offer single-cell resolutions, providing essential insights into cellular organizations and dynamics. Yet, the complexity of such data presents analytical challenges and demands substantial computing resources. Moreover, the proliferation of diverse spatial omics technologies, such as Xenium, MERSCOPE, CosMX in spatial-transcriptomics, and MACSima and PhenoCycler in multiplex imaging, hinders the generality of existing tools. We introduce Sopa (https://github. com/gustaveroussy/sopa), a technology-invariant, memory-efficient pipeline with a unified visualizer for all image-based spatial omics. Built upon the universal SpatialData framework, Sopa optimizes tasks like segmentation, transcript/channel aggregation, annotation, and geometric/spatial analysis. Its output includes user-friendly web reports and visualizer files, as well as comprehensive data files for in-depth analysis. Overall, Sopa represents a significant step toward unifying spatial data analysis, enabling a more comprehensive understanding of cellular interactions and tissue organization in biological systems.

Spatial omics data offer opportunities to improve our understanding of cellular interactions within their micro-environment and the intricacies of tissue organization[1,2]. Recent advancements in imaging technologies have expanded these capabilities, enabling the measurement of 1000+ genes through Spatial Transcriptomics[3] and/or the analysis of 50+ proteins via Multiplex Imaging[4]. These include Merfish[5], ISH[6], ISS[7], MICS[8], PhenoCycler[9] and IMC[10], all of which provide single-cell resolution that could not be achieved by previous spot-based techniques like 10× Visium or Nanostring GeoMX[11]. Therefore, image-based technologies provide a higher resolution—up to the subcellular level—which is needed for a detailed exploration of individual cells and their gene expression profiles within their spatial context. This level of precision has been essential for unravelling tissue architecture and understanding cellular interactions; it marks the beginning of a significant leap forward in our comprehension of biological systems[9,12,13].

In parallel with these technological advancements, the analysis of image-based spatial omics has encountered significant computational challenges and limitations[3,14–17]. Most existing methods[18–20] are not designed to handle large images with millions of cells. Their usage typically demands high-performance computational clusters with substantial memory resources, which limits accessibility to spatial omics due to cost and hardware constraints. As a result, most companies have developed proprietary tools for their own data types, primarily focusing only on segmentation and visualization. Yet, these proprietary tools have certain constraints, such as (i) a limit on specific functionalities, (ii) no incorporation of the latest state-of-the-art methods, and (iii) a lack of versatility, as they cannot be applied to other technologies. This tool diversity has other limitations in that each suite has a learning and adaptation process and that the tools' specificities lead to variations in the analysis of comparable data types.

[1]Paris-Saclay University, CentraleSupélec, Laboratory of Mathematics and Computer Science (MICS), Gif-sur-Yvette, France. [2]Paris-Saclay University, Gustave Roussy, Villejuif, France. [3]Gustave Roussy, Department of Medical Oncology, Villejuif, France. [4]These authors contributed equally: Quentin Blampey, Kevin Mulder. ✉e-mail: quentin.blampey@gmail.com; paul-henry.cournede@centralesupelec.fr

Similarly, current open-source analysis libraries often rely on (i) already-segmented data[21,22], (ii) specific data types[23,24], or (iii) a subset of analysis tasks[23,24], resulting in fragmented approaches and difficulty in adapting one approach to a different type of technology. The absence of a unified data representation and modular programming interface further complicates the integration of various analysis steps.

To address these gaps, our work introduces Spatial Omics Pipeline and Analysis, or Sopa, a computational framework that enhances the accessibility, efficiency, and interpretability of image-based spatial omics data. Sopa is a memory-efficient pipeline that works across all image-based spatial omics technologies and that can display results in a common visualizer. This includes the most recent Spatial Transcriptomics technologies (Xenium, MERSCOPE, CosMX) and also the multiplex imaging techniques (e.g., MACSima, PhenoCycler, Hyperion). Sopa's capabilities include segmentation and multilevel annotation, both based on transcripts and/or stainings, as well as spatial statistics and niche geometry analysis. We demonstrate Sopa's performance on four public datasets: two spatial-transcriptomics (MERSCOPE, Xenium) and two multiplex imaging technologies (PhenoCycler, MACSima), and provide a memory and time benchmark over multiple dataset sizes. Additionally, we demonstrate Sopa's capabilities for geometric and spatial analysis on the MERSCOPE dataset by analyzing cell colocalization with regard to cell types and niches, showing promise for biological discoveries. All these functionalities

are accessible via our open-source code, which includes a Command Line Interface (CLI), an Application Programming Interface (API), and a flexible Snakemake[25] workflow, enabling users with various levels of expertise to process spatial omics data seamlessly, from no-code simplicity to full flexibility. The pipeline's generic nature ensures effortless transitions to other types of spatial omics data, making it a versatile and powerful tool for the scientific community.

## Results

### Technology-invariant pipeline for spatial omics

To establish versatile tools, a common strategy involves adopting a shared data structure that seamlessly integrates across diverse technologies. SpatialData[26] serves as one such comprehensive framework, including readers tailored for the most widely used spatial omics technologies. Building upon this, Sopa converts any data into a SpatialData object, on which all of the six following tasks are performed. First, if needed, users can interactively select a region of interest, facilitating the exclusion of less relevant or lower-quality areas. Next, we generate overlapping patches of images and/or transcripts. Segmentation can then be performed for each individual patch, and we currently support Cellpose[18] (image-based segmentation) and Baysor[19] (transcripts-based segmentation). Afterwards, the cell segmentation masks are converted into polygons and merged over all patches to remove potential artefacts. Following these first four steps, we average

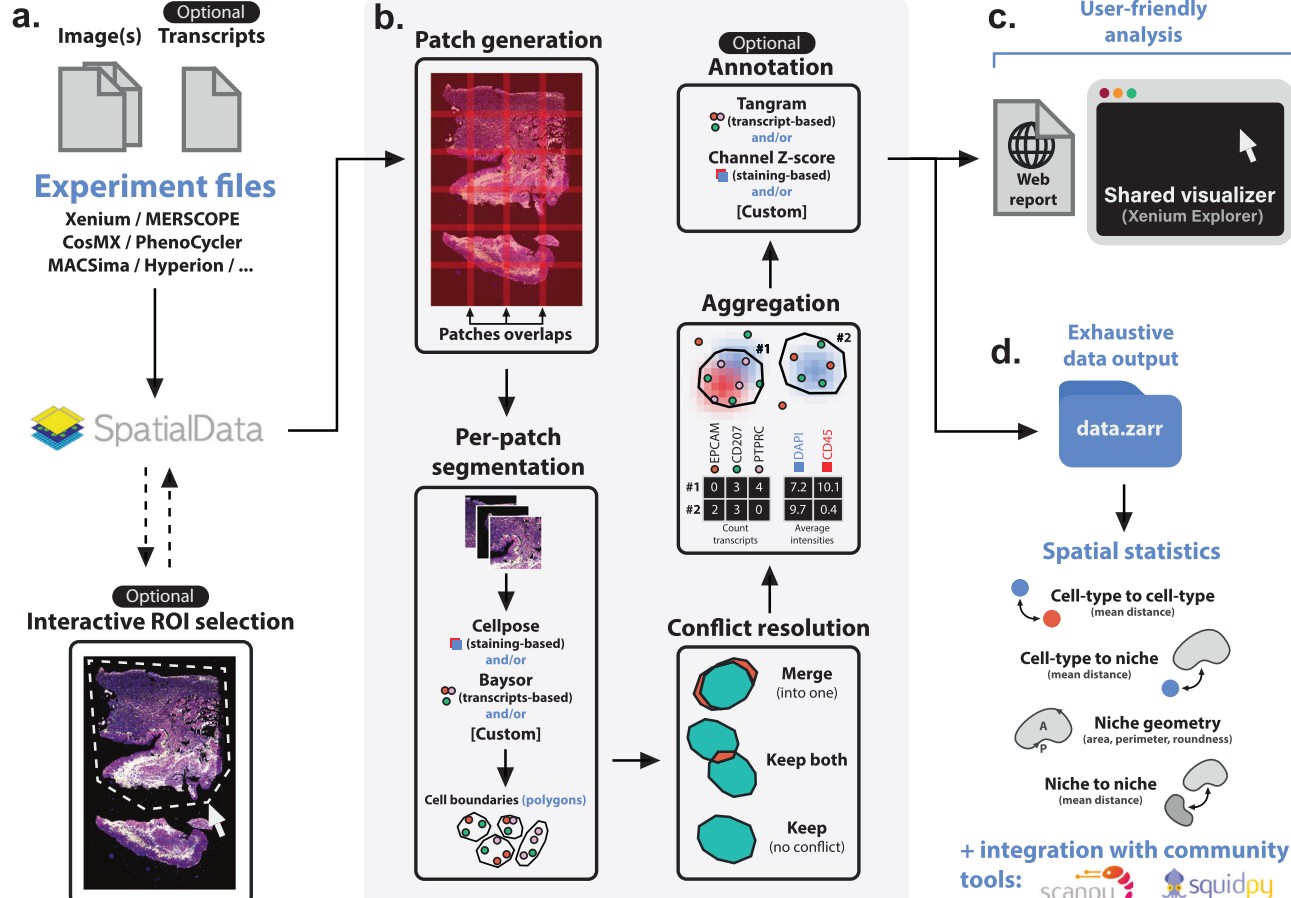

**Fig. 1 | Overview of Sopa. a** The pipeline input consists of experimental files of any image-based spatial omics. It is transformed into a SpatialData object, on which we can optionally select a region of interest (ROI) interactively. **b** Afterwards, the data is split into overlapping patches, and segmentation is run on each patch (for instance, Cellpose, Baysor, or a custom segmentation tool). Since patches are overlapping, some cells can be segmented multiple times on different patches. Therefore, these conflicts have to be resolved: two boundaries with a significant overlap are merged into one cell, while two cells barely touching are kept separate. The next step is aggregation, i.e., counting transcripts and averaging each channel intensity inside each cell. This allows annotation, either based on transcripts (using Tangram) or on channel intensities. **c** Afterwards, Sopa outputs a user-friendly report and files to be opened in the Xenium Explorer (whatever the input technology). **d** All data files are kept for further analysis in Sopa, such as spatial statistics, or integration with community tools.

the staining intensities and count the transcripts inside each cell, allowing further tasks such as annotation. For example, Sopa currently supports Tangram[20] for transcript-based annotation, and a simple Z-score method for staining-based annotation. Finally, we implemented spatial and geometric analysis tools to fully exploit the spatial nature of the data. For convenience, all image-based technologies can be visualized in a shared explorer, and an HTML report is provided for pipeline quality checks. The full process described above is summarised in Fig. 1.

## Visualisation of spatial omics in a cross-technology interactive visualizer

In spatial omics analysis, effective visualization is crucial but has presented challenges due to the size of the datasets. While open-source initiatives like Napari[27] are emerging, they currently face limitations in handling large amounts of transcripts. Also, most companies provide technology-specific visualizers, offering limited user possibilities (see Supplementary Notes). Yet, 10× Genomics has introduced the Xenium Explorer, an optimized visualizer whose file format is open, i.e., formats that can be generated for various SpatialData types. In Sopa, we have incorporated a converter that transforms the pipeline output into the input files compatible with the Xenium Explorer (Fig. 1c). This integration ensures access to an efficient and robust visualizer, extending its functionalities to any technology whose data is readable by Sopa. Importantly, this adaptation applies to both spatial transcriptomics and multiplex imaging data, with the "Transcripts" panel selectively available for transcriptomics data. Fig. 3b, e shows views using this Explorer, while Supplementary Fig. 2, 3 provide full-window examples. In addition to visualisation, the Xenium Explorer contains an interactive tool to align images from which we can export a transformation matrix and use it to align images on the SpatialData object to benefit from all the functionalities in Sopa (see Supplementary Notes).

## Memory and time efficient analysis of spatial omics

Managing large datasets is a critical challenge in spatial omics, particularly when dealing with images that can reach hundreds of gigabytes and contain hundreds of millions of transcripts in spatial transcriptomics data. This necessitates implementing memory optimization techniques to ensure the scalability of the analysis. Notably, segmentation algorithms like Cellpose[18] and Baysor[19] encounter scalability issues with large images, as illustrated in Fig. 2a, b. To tackle this, these segmentation models are applied to smaller regions called patches, drastically decreasing random-access-memory (RAM) usage and time. While this patching process generates possible segmentation conflicts, we show in Fig. 2d, e and in Supplementary Fig. 1 that this does not impact segmentation quality, since most conflicting cell boundaries have an intersection-over-min-area (IOMA) lower than 0.07 or higher than 0.8 (see Supplementary Notes and Supplementary Table 1 for more details). Indeed, for cells on overlapping regions, most of the boundary conflicts correspond to either (i) the same cell segmented twice on the two patches (at least one cell is complete, as shown in Fig. 2c, with one boundary being included in the other), or (ii) different cells slightly overlapping (as shown in the right of Fig. 2c). Additionally, the conventional storage of cell boundaries as raster masks demands significant memory for storage and processing (see Fig. 2f). To address this, we adopt a more efficient approach by storing cell boundaries as polygons using Shapely[28], which proves highly effective for both on-disk and in-memory storage. This also facilitates geometry-related operations, such as cell expansion, area/perimeter computations, and cell-cell intersections. Combined with the image lazy loading feature from SpatialData[26] and Xarray[29], we implement a fast channel averaging on cell boundaries by combining geometry operations and image chunk lazy loading (see Fig. 2f), i.e., deferring memory loading until needed for processing. Additionally, using

memory-efficient tools like Dask[30], we extend geometric operations of GeoPandas[31] on chunks of transcripts, ensuring parallel processing of as many chunks as possible without exceeding memory limits (see Fig. 2g). For image conversion to a pyramidal '.tif`, we significantly lower the memory usage compared to what is recommended by 10X Genomics by writing tiles in a lazy manner, which avoids loading the full image in memory (see Fig. 2h). To highlight Sopa's memory efficiency, we compared its RAM usage against standard methods for all tasks mentioned above across various dataset sizes, summarized in Fig. 2. Overall, the latter figure shows significant improvements in terms of RAM and time: depending on the tasks, Sopa can require between 10 and 100 times less memory than normal techniques and can be up to 100 times faster. Even on the largest image, Sopa can be run with a simple laptop with 16GB of RAM.

## A wide range of use cases for different levels of expertise

Sopa offers three distinct options, each tailored to different use cases: (i) a Snakemake[25] pipeline that enables a quick start within minutes, (ii) a CLI that facilitates rapid prototyping of a personalized pipeline, and (iii) an API that allows direct usage of Sopa as a Python package (https://github.com/gustaveroussy/sopa), providing full flexibility and customization. The Snakemake pipeline remains consistent across various technologies, with only its configuration differing. Users can leverage existing configuration files, selecting one that aligns with their technology, which then enables them to execute the pipeline without any code updates. Another advantage of Sopa's generality and scalability is that more advanced users seeking customisable pipelines can use the CLI. Notably, Sopa's general design allows for an easy integration of any state-of-the-art or custom segmentation methods such as BIDCell[32], rendering them memory-efficient and accessible for all image-based spatial omics applications. Additionally, the Python API is available for users interested in incorporating specific parts of Sopa into their personal libraries. This API also facilitates integration with other tools of the scverse[33] ecosystem, such as Scanpy[34] or Squidpy[22] (see Supplementary Notes). In particular, the integration with Squidpy enables the use of post-processing tools for cell-cell interaction and spatially variable gene analysis.

## High resolution of the tumour microenvironment

Segmentation plays a crucial role in image-based spatial omics analysis. Sopa focuses significantly on improving this step by enabling the usage of state-of-the-art segmentation models like Baysor[19] on large datasets. Indeed, as shown in Fig. 2a, b, these high-quality segmentation tools use a lot of memory, which hinders their usage on large spatial datasets. To evaluate the resolution provided by Sopa after segmentation, we annotated major cell types and conducted tests on four datasets: two spatial-transcriptomics datasets (MERSCOPE and Xenium) and two multiplex-imaging datasets (PhenoCycler and MAC-Sima), Supplementary Notes for more details. For the MERSCOPE and Xenium datasets, proprietary segmentations were provided by Vizgen and 10X Genomics, respectively. In comparison to these segmentations, Sopa shows an improved cell-type distinction on UMAP[35] plots (see Fig. 3a, d) by leveraging Baysor. To support these visual observations, we used multiple metrics, indicating that Sopa can generate more significant population-specific genes, greater intra-cluster distance, and improved cluster separation (see Fig. 3c, f). The increased resolution in spatial omics data allows for a more in-depth exploration compared to previous segmentations (see Supplementary Fig. 4 for more details).

Sopa also facilitates the concurrent analysis of both RNA and proteins. To demonstrate this, we used the Xenium dataset, which includes transcriptomic expression and protein stainings (CD20, PPY and TROP2). CD20 is a common marker for B cells, PPY is expressed by endocrine cells, and TROP2 is overexpressed in tumour cells. 10× Genomics currently does not produce files with protein expression per

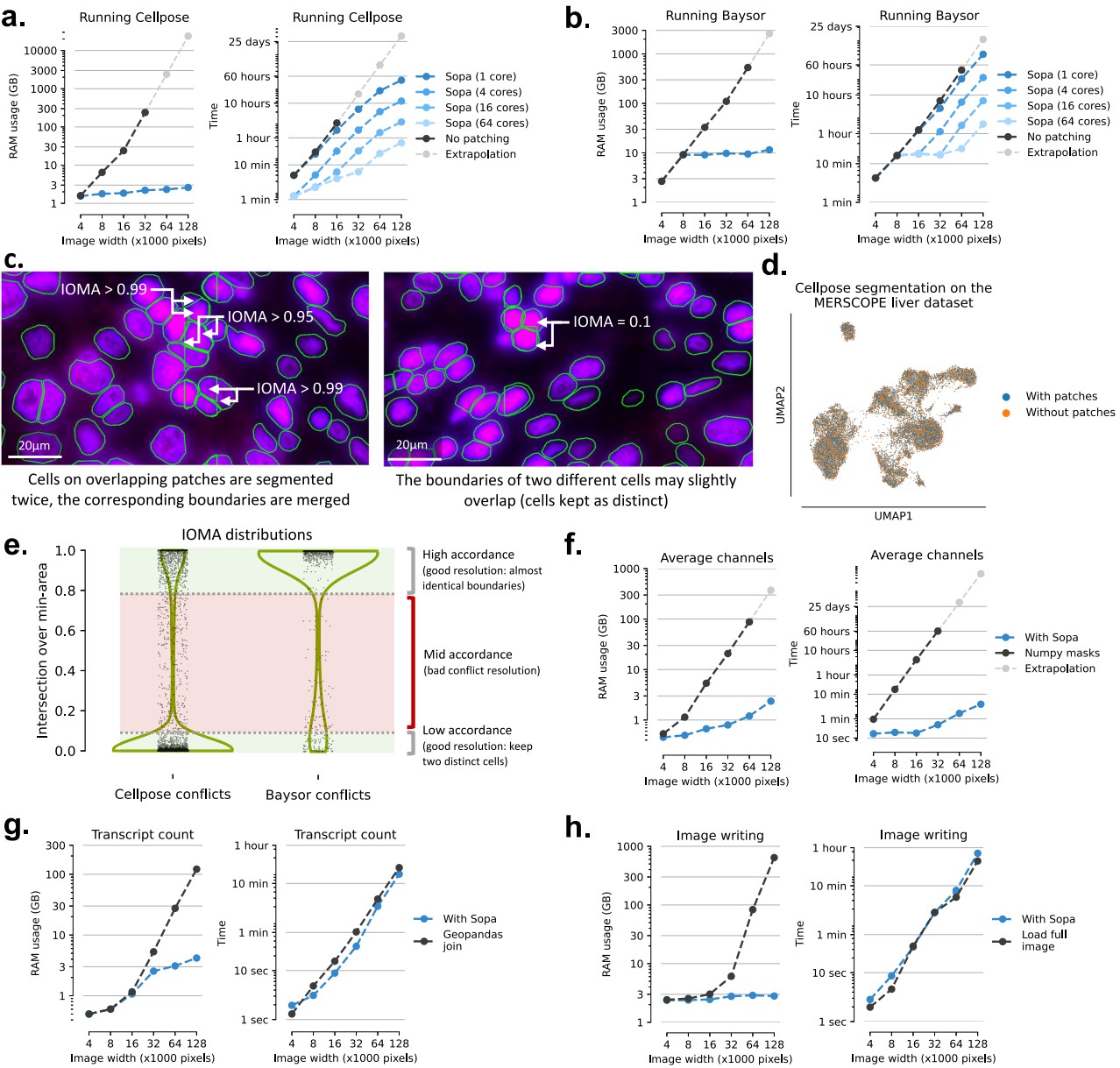

**Fig. 2 | Computational efficiency of Sopa in terms of RAM and time on different dataset sizes. a** Cellpose segmentation comparison: with and without patching. The RAM usage is given per core. **b** Baysor segmentation comparison: with and without patching. The RAM usage is given per core. **c** Examples of cell boundaries before resolving the conflicts over overlapping patches when running Cellpose segmentation on DAPI staining (MERSCOPE human liver hepatocellular carcinoma dataset). On overlapping regions, cells are segmented twice (middle and right). For each conflict, their IOMA determines whether or not to merge the two cell boundaries. **d** UMAP showing the difference between the resolution with and without the patching process. **e** Violin plots showing the intersection-over-min-area density of segmentation conflicts when using patches (for both Cellpose and Baysor). When resolving a conflict, the two good cases are either (i) a high concordance between the two cells (which will be merged), or (ii) a low concordance between them (the two cells are kept). IOMA below 0.07 or above 0.8 correspond to good conflict resolution cases. **f** Channels averaging for each cell: Sopa and standard average inside numpy masks. **g** Counting each gene inside each cell: with Sopa compared to GeoPandas join operation on the whole DataFrame. **h** Writing image as a tiff file for the Xenium Explorer: with Sopa compared to what is recommended by 10× Genomics, i.e., loading the whole image in memory. Source data are provided as a Source Data file.

cell, while Sopa does support the analysis of proteins. To demonstrate this feature, we aligned the Xenium staining image to the original coordinate system (see Supplementary Notes), and Sopa computed the CD20/PPY/TROP2 intensity within all cell boundaries. Combined with transcriptomic expression, CD20 staining greatly facilitates the annotation of B cells, as shown by their clear delimitation in Fig. 3d and Supplementary Fig. 4c. In the future, we expect technologies to be able to run more protein stainings in parallel with transcriptomics data, making this kind of analysis even more valuable.

Regarding multiplex imaging, Sopa shows efficiency in (i) managing large protein staining panels and (ii) segmenting millions of cells (using Cellpose). The former is exemplified by the MACSima dataset with 61 stained proteins. Again, we computed staining intensity per cell, and Fig. 3g demonstrates Sopa's capacity to annotate high-resolution cell types. Secondly, the PhenoCycler dataset underscores Sopa's ability to handle datasets of substantial size, with an area of 3 cm², containing approximately 2,500,000 cells. The corresponding cell resolution is shown in Fig. 3h, i.

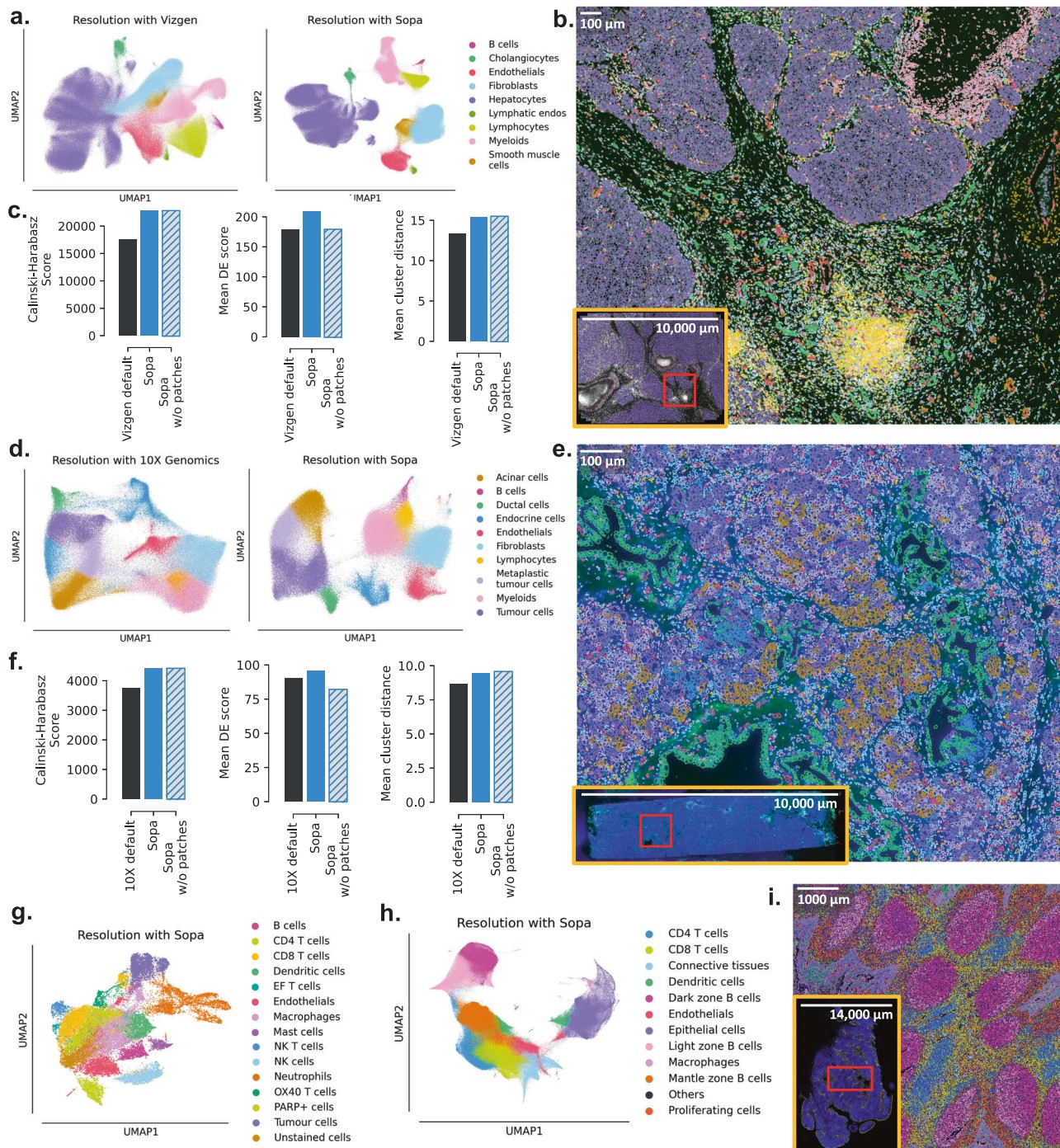

**Fig. 3 | Data resolution after Sopa segmentation over two spatial-transcriptomics technologies (MERSCOPE (a–c) and Xenium (d–f)) and over two multiplex-imaging technologies (g–i). a** UMAPs after Vizgen proprietary segmentation on the MERSCOPE human liver hepatocellular carcinoma dataset (left) and after Sopa segmentation on the same dataset (right). **b** Visualization of cell types on the MERSCOPE dataset after annotation with Sopa. Colours correspond to the legend of **a. c** Three cluster separation metrics compare the quality of these two segmentations on the MERSCOPE dataset. The grey hatched boxes extrapolate the score Sopa would have without running on patches. **d** UMAPs of cells after 10× Genomics proprietary segmentation on the Xenium human

pancreatic cancer dataset (left) and after Sopa segmentation on the same dataset (right). **e** Visualization of cell types on the Xenium dataset after annotation with Sopa. Colours correspond to the legend of **d. f** Three cluster separation metrics compare the quality of these two segmentations on the Xenium dataset. The grey hatched boxes extrapolate the score Sopa would have without running on patches. **g** UMAP of cell types on the MACSima dataset (head and neck squamous cell carcinoma), based on 61 protein stainings. **h** UMAP of cell types on the PhenoCycler dataset (human tonsil), based on 31 protein stainings. **i** Cells of the PhenoCycler dataset visualized. The colours correspond to the legend of **h**. Source data for **c**, **f** are provided as a Source Data file.

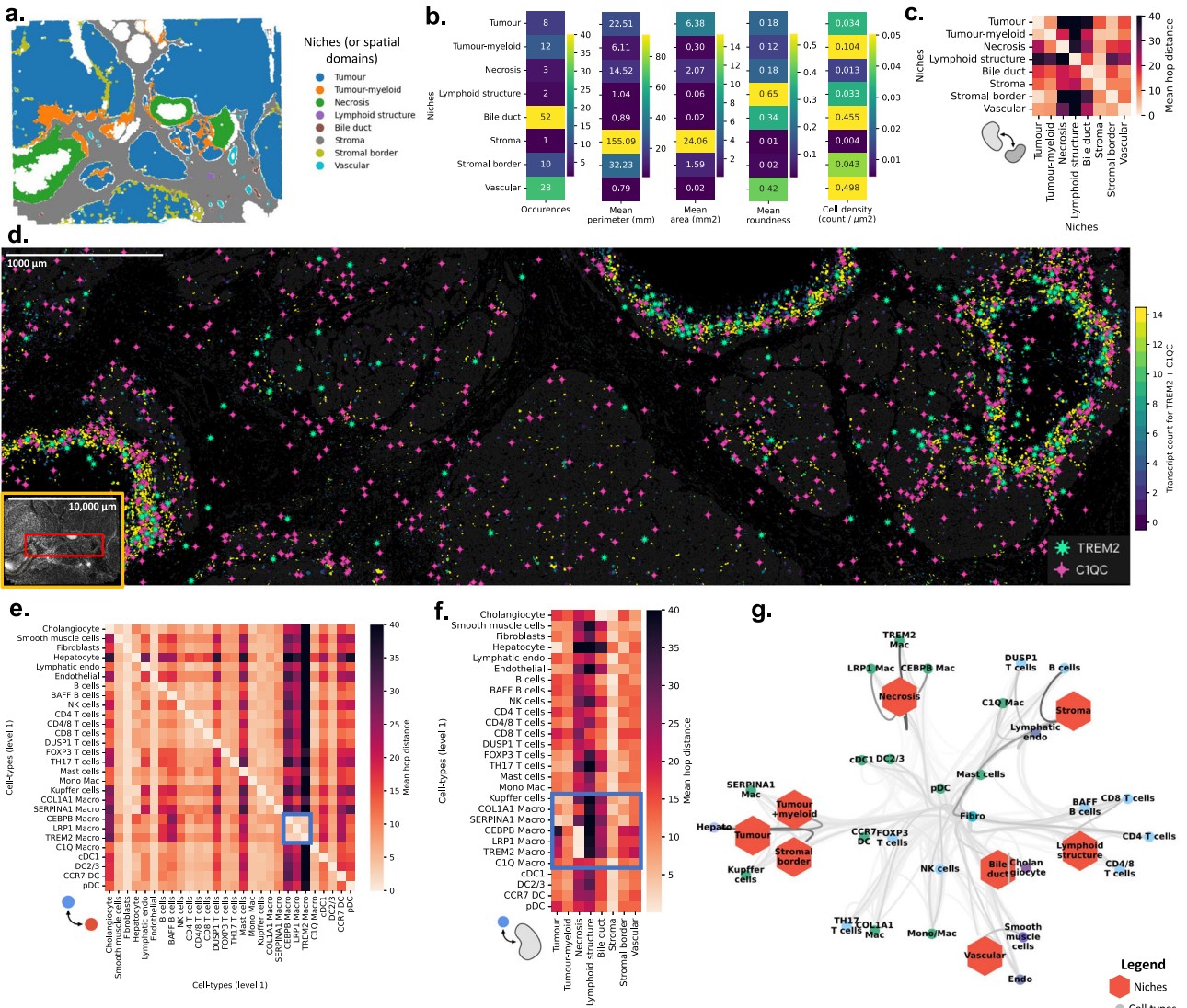

**Fig. 4 | Geometric analyses and spatial statistics on the MERSCOPE human liver hepatocellular carcinoma dataset. a** Niches (or spatial domains) after geometric conversion to shapely polygons. **b** Geometric statistics of the niches: their occurrence, perimeter, area, roundness, and inner cell density. **c** Heatmap of average hop distance between niches and niches. **d** Localisation of *TREM2* macrophages shown in the visualizer. The *TREM2* and *C1QC* genes are shown, and cells are coloured by their gene counts for the two selected genes. **e** Heatmap of average hop distance between cell types and all other cell types. *LRP1*, *CEBP*, and *TREM2* macrophages show a high proximity. **f** Heatmap of average hop distance between cell types and

niches. The macrophage subpopulations show heterogeneous localisation with respect to the niches. *LRP1*, *CEBP*, and *TREM2* macrophages are enriched in the necrosis niche. **g** Network plot summarising the distance metrics of **c**, **e**, **f**. Each node of the network corresponds either to a niche (hexagon) or a cell type (circle). The lower the mean distance between the two nodes, the higher the weight of the edge between these two nodes. A high node-node proximity is shown by a dark edge. Overall, it provides an overview of the colocalisation of cell types and niches in the tumour environment.

In summary, these studies demonstrate that Sopa can (i) be applied across diverse technologies, (ii) efficiently handle millions of cells, and (iii) seamlessly operate on both transcriptomics and protein stainings.

**Demonstration of geometric and spatial analyses capabilities**

Spatial omics naturally unlocks multiple biological questions related to spatial organization. While some are addressed in libraries such as Squidpy[22], metrics related to the distance between cell-types/niches and the geometric characteristics of those niches are not provided. These metrics could help in the understanding of the morphology of the tumour micro-environment and its location with regard to different cell types. Such statistics have been shown to be relevant for predicting disease progression or response to treatment[36,37]. For instance, it is known that tertiary lymphoid structures (TLS) have a

good prognosis[38], but their geometry has not been studied. TLS may come in different sizes, shapes, occurrences, or locations with regard to other niches. Such statistics are generalized for all cell categories (usually, cell types or niches). Leveraging this spatial analysis, we demonstrate a better understanding of the dynamics among different cell types and their relation to different spatial niches on the MERSCOPE liver dataset (Fig. 4). To use Sopa geometric analysis, we run STAGATE[39] to identify eight distinct niches (or "spatial domains") across various tumour regions (Fig. 4a). First, we show in Fig. 4b four geometric properties related to these niches: for each niche compartment, we counted their occurrence on the same slide, as well as their mean area, perimeter, and roundness. For instance, our geometric analysis shows a high occurrence of vascular niches, that are small in area and perimeter, but have a high roundness. Conversely, the stroma has only one occurrence and is highly "unround", and

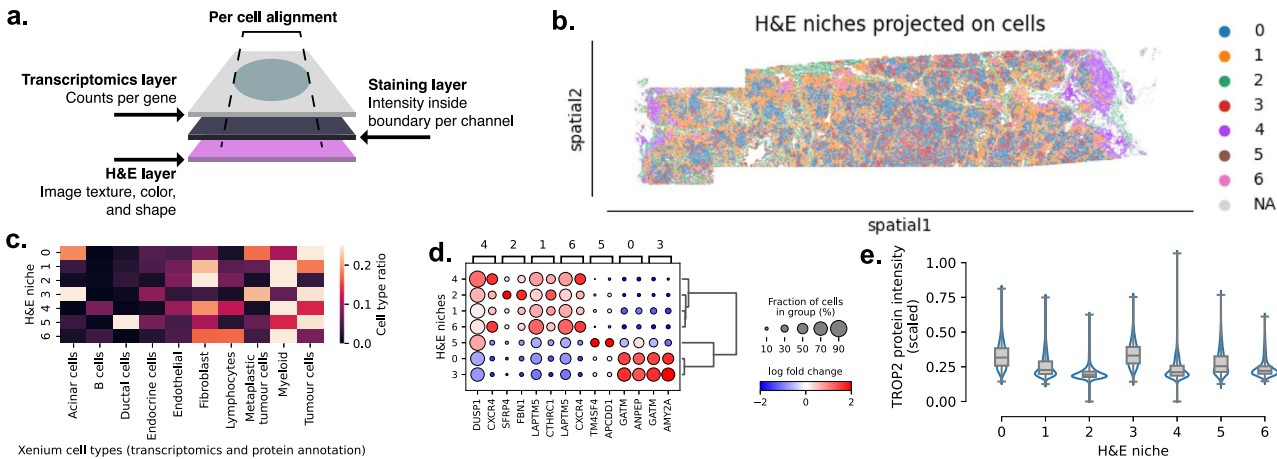

**Fig. 5 | Spatial multi-omics analyses on the Xenium pancreatic cancer dataset.**
**a** Overview of spatial multi-omics alignment. For each single cell, the information of (i) transcriptomics, (ii) stainings, and (iii) H&E is combined after the alignment of all the different layers. **b** H&E clusters of patch-level embeddings based on a pre-trained computer vision model (denoted as H&E niches). The figure shows the cells obtained from spatial transcriptomics data and coloured by the H&E patch cluster inside which they are included. **c** Proportion of cell types inside each H&E niche. The cell types are the cell types annotated using both spatial transcriptomics and protein information as in Fig. 3. **d** Differential gene expression performed on the H&E niches using single-cell resolution. **e** Distribution of TROP2 intensities per cell ($N = 175,022$) inside each H&E niche, showcasing the usage of the staining layer coupled with the H&E information. Source data for **e** are provided as a Source Data file.

Fig. 4c shows that this shape enables a "proximity" to every other niche. Fig. 4c also highlights how far the vascular niche is from the necrosis. While such observations are not novel, our geometric computation allows for statistical comparisons over multiple patients, which could lead to the discovery of significant geometric biomarkers in large-scale studies. Finally, Squidpy[22] already incorporates functionalities on neighbourhood enrichments, which is a local metric and, therefore, not suited to capture niche-level information. In comparison, the distance metric used in Sopa can capture asymmetrical observations and global organizations (see Supplementary Fig. 6 for more details).

We also utilised Sopa to assess the intricacies of the tumour complexity. We annotated the immune populations of the MERSCOPE dataset in higher definition (see Supplementary Fig. 5a, b) and, in parallel, performed a differential analysis on each niche to better understand niche complexity. This revealed a distinct necrotic niche correlated with *TREM2* macrophages (expressing *TREM2*, *C1QC*, and *CSF1R*), a population of macrophages reported across cancer types and often associated with bad prognosis[40,41] (see Fig. 4d and Supplementary Fig. 5c). To deepen this understanding of tissue intricacies, we investigated whether these *TREM2* macrophages were in close distance with any other cell type (see Fig. 4e). Strikingly, this figure highlighted that three macrophage populations (*LRP1*, *CEBP*, and *TREM2*-macrophages) exclusively interacted with themselves. Correlating their location with the niche revealed that their co-occurrence is specific to the necrotic niche (see Fig. 4f). When combining all (cell-cell/cell-niche/niche-niche) interactions, this affirms again the association of *LRP1/CEBP/TREM2*-macrophages in the necrotic niche, yet it also highlights the heterogeneity of all macrophage populations and their relation to the niche in the whole tissue environment (see Fig. 4g). These combined interactions also showed that, inversely, the conventional dendritic cells (DCs) are not associated with any niche environment, accentuating how some populations can also be niche-independent. This observed spatial location underscores a potential reprogramming feature of macrophages based on their specific niche. While it is known that the accumulation of *TREM2* macrophages has been associated with enrichment in the tumour regions[42–44], Sopa can provide insights for a refined understanding of a macrophage-specific tumour-associated phenotype. These examples illustrate that this geometric and spatial analysis—computed with Sopa—helps better

understand the tumour's architecture and its relationship with cell type phenotypes.

### Incorporation of H&E into the multi-omics spatial analysis

Some technologies, such as the Xenium, have been developed to get Hematoxylin and Eosin (H&E) staining and protein staining on the same slide used for Spatial Transcriptomics. By aligning the modalities (with the Xenium Explorer, as detailed in the Supplementary Notes), Sopa enables analyses that can interplay with all three modalities. Especially, the H&E modality, via the colour and texture, captures extra information that the two other modalities do not contain. For instance, H&E may be stronger in regions with low RNA information, such as high collagen regions (see Supplementary Fig. 7). In Fig. 5, we perform analyses that couple the three layers to provide interpretability to the H&E niches. Fig. 5c shows that H&E-based niches are highly heterogeneous in terms of cell types, with some niches being highly enriched in some particular populations. Notably, niche 3 is highly specific to Acinar cells, niche 5 is specific to Ductal cells, while niche 4 is enriched in B cells and Myeloid cells. Also, Fig. 5d shows differentially expressed genes inside each niche, providing complementary insights to Fig. 5c, such as *TM4SF4* and *APCDD1* being highly specific to niche 5. Finally, Fig. 5e exemplifies the analysis of the distribution of protein stainings inside these H&E niches, with TROP2 being more expressed in niche 0 and 3, which correspond to the tumour-specific niches identified in Fig. 5c. Overall, these examples show the capability of Sopa to use one spatial modality to bring insights into another spatial modality.

## Discussion

Advances in technology development for spatial omics hold great promise for biological discoveries. Yet, to build strong and unified foundations for spatial omics data analysis, more tools are required. With this purpose in mind, we designed and built Sopa to address several crucial aspects of spatial omics analysis: versatility, reproducibility, and scalability. It offers a suite of tools—or building blocks—designed for spatial omics, which are assembled to build a pipeline for any image-based spatial omics technology. At the end of the pipeline, it produces standardized outputs, which ease exploration and visualization. While each company's technology comes with its own suite of tools—which differ in terms of capabilities and functionalities—Sopa does not require learning from multiple data types and software.

In addition, Sopa is scalable from simple laptops to high-performance clusters, offering further versatility for users.

Moreover, Sopa can easily integrate recent methods and tools: as future segmentation or annotation methods are developed, they can be added to Sopa once published and validated. This incorporation into Sopa enables scalability and availability to any future technology with only minor configuration changes. As datasets become increasingly bigger, Sopa's scalability is crucial. For instance, Sopa enabled the possibility of running Baysor on data produced by the MERSCOPE, which was previously impossible due to RAM usage and time. Assessing the effect of patch-based segmentation showed no significant difference in segmentation quality. We also demonstrated that Baysor significantly increases data quality compared to the default Vizgen and 10X Genomics segmentation tools, which aligns with Hartman et al.[45].

As shown on the MERSCOPE liver dataset, we were able to annotate spatial-specific macrophages, particularly *TREM2* macrophages, in the necrotic niche. Additionally, *TREM2* has been shown to increase with HCC, suggesting a potential immunosuppressive role of *TREM2*[42,44], while necrosis has been associated with worse prognosis[46,47]. With the help of Sopa, the exploration of this relationship between tissue architecture and cell phenotypes can advance biological knowledge.

Besides higher data resolution, Sopa can also incorporate protein and H&E information into spatial analysis. Without this protein layer, extracting the B cell population in the Xenium data would not have been possible. Concerning the H&E layer, we can benefit from the transcriptomics layer to bring interpretability to the H&E tissue characterization or also build upon Sopa to develop tools that predict refined spatial-transcriptomics cell types based on H&E images. While current spatial technologies involve either a high number of proteins or transcripts, future developments could add extra layers of information, contributing to a better understanding of biological systems. This paper has demonstrated through various techniques that Sopa is ready to handle large multimodal spatial technologies.

## Methods

### Datasets used

Four public datasets were used to demonstrate Sopa's abilities. First, we used a MERSCOPE dataset (from Vizgen) of the human liver hepatocellular carcinoma (HCC), called FFPE Human Immuno-oncology Data Set May 2022. It is composed of a 500-gene panel, and has DAPI staining and PolyT staining. It contains about 500,000 cells, depending on the segmentation. Secondly, we used a Xenium dataset (from 10X Genomics) of pancreatic cancer (adenocarcinoma, Grade I-II) with the Xenium Human Multi-Tissue and Cancer Panel, in parallel with corresponding H&E image, and a protein-staining image with DAPI/CD20/PPY/TROP2. Note that the two latter images has to be aligned on the default main DAPI image. It contains about 180,000 cells, depending on the segmentation. Thirdly, we used a PhenoCycler dataset (from Akoya Biosciences) of the human tonsil (FFPE) with 31 protein stainings. It contains about 2,500,000 cells, depending on the segmentation. Finally, we used a MACSima dataset (from Miltenyi) of head and neck squamous cell carcinoma (HNSCC) with 61 protein stainings. It contains about 40,000 cells, depending on the segmentation. For more details about the accessibility of these datasets.

### Metrics used and computational details

The Calinski-Harabasz-Score is defined as the ratio of the sum of between-cluster dispersion and of within-cluster dispersion. To compute this score, we used the implementation in scikit-learn[48]. The mean cluster distance is the average distance between all pairwise combinations of cells between two different clusters; thus, a higher distance

indicates a better cluster separation. For the differential expression analysis, we ran the scanpy[34] *rank_genes_groups* function, and we averaged the score of the 20 most significant genes for each cell type. Since we could not run Baysor on the full datasets in Fig. 2, we run it on 16,000-pixels-wide crops of the MERSCOPE and Xenium datasets, and we computed the ratios between the run with the patches and without patches. We then averaged these ratios across these two datasets, with two runs on each dataset, for each of the above metrics and used the resulting ratios to extrapolate the Baysor score on the full datasets. The time and memory benchmarks were performed on a Slurm cluster on the same CPU nodes. The benchmark related to Cellpose was performed on crops of the MERSCOPE dataset, while the other time and memory benchmarks were performed on a synthetic dataset (see Supplementary Notes). Fig. 2e was generated based on the corresponding 16,000-pixels-wide datasets; this involves 25 Cellpose patches and 4 Baysor patches. The percentage of conflicts for Cellpose (compared to all pairs of cells) was 0.007%, while this percentage was 0.001% for Baysor. The UMAPs of Fig. 3 were generated with scanpy[34], using the default parameters. The MERSCOPE and Xenium datasets have been segmented with Baysor, while the PhenoCycler and MACSima datasets have been segmented with Cellpose. Both the MERSCOPE and Xenium datasets have been annotated using Tangram (see Supplementary Notes for more details). Concerning the H&E niches, they were obtained by running a ResNet[49] model pre-trained on ImageNet and applied on patches of size 250 × 250 pixels.

### Segmentation on patches

For computational efficiency, segmentation is performed on patches, i.e., small image regions. These patches have a certain overlap, which is typically chosen to be at least twice as big as the average diameter of cells (e.g., 20 microns). This way, each cell should be complete in at least one patch, which avoids artefacts due to cutting cells at the border of the patches. Subsequently, any segmentation algorithm compatible with images and/or transcripts can be applied. While Cellpose[18] and/or Baysor[19] are commonly used, Sopa does allow the integration of other segmentation algorithms. Following segmentation on individual tiles, the cell boundaries are transformed into polygons using Shapely[28]. Since patches overlap, some cells may be segmented across different patches, leading to segmentation conflicts where multiple polygons correspond to a single cell. To resolve this, we adopt a method similar to the one used in Vizgen's preprocessing tool (VPT). Specifically, we merge pairs of cells when the intersection area exceeds half the area of the smaller cell, ensuring a substantial overlap. If the intersection area is too small, indicating distinct cells, both polygons are retained. When the overlap area divided by the smallest cell area is close to 1, this corresponds to two almost identical cells, while a score close to 0 corresponds to two cells barely touching. In Fig. 2e, we studied the distribution of this score, showing that most of the conflicts are associated with a score that is either very close to 0 or very close to 1, indicating a good conflict resolution. Indeed, statistical considerations indicate that scores above 0.8 or below 0.07 are good resolutions (see Supplementary Notes). Additionally, note that, before segmentation, the user can decide to select a region of interest: this can be done interactively with matplotlib[50] on a low-resolution image.

### Channel averaging

When dealing with image-based technologies, a crucial step involves averaging the intensity of each channel within each cell. While this task can be achieved using cell masks, it proves highly inefficient in terms of both time and memory consumption. To address this challenge, we adopt a chunk-level approach: (i) For each chunk, we identify cell boundaries (i.e., polygons) that intersect with the chunk coordinates, then (ii) we determine the bounding box for each of these cells, then

(iii) we extract the image values for each of these bounding boxes, and finally (iv) we rasterize the cell polygons to average the staining intensity over the local bounding box. In this way, we only load small arrays corresponding to each cell, instead of loading large cell masks. This process is repeated over all chunks, and we make sure that the channel intensity for cells located on multiple chunks is computed correctly.

## Counting transcripts

GeoPandas[31] is a Python library that enhances Pandas[51] Dataframes by incorporating support for Shapely[28] geometries. It facilitates scaling operations on geometries, making it particularly suitable for transcript counting, where transcripts can be represented as Shapely points and cells as Shapely polygons. However, without Sopa, the memory requirements for such operations can be substantial, especially for spatial transcriptomics datasets that may contain up to one billion transcripts. To optimize this process, we leverage Dask and execute the GeoPandas[31] "join" operation at the partition level to assign each point (i.e., a transcript) to a polygon (i.e., a cell). Thus, each operation is carried out on smaller data frames, each less than 100MB in size. Dask efficiently assigns each partition to different workers in parallel, mitigating memory concerns. This approach proves highly effective on both laptops and high-performance clusters, as Dask is designed to seamlessly scale these processes without necessitating any code modifications.

## Conversion to the Xenium Explorer

Converting a spatial omics object into the Xenium Explorer requires the creation of six files: (i) the image, (ii) a JSON metadata file, (iii) the cell boundaries, (iv) the cell categories (e.g., cell type or clustering), (v) the gene counts table, and (vi) the transcripts (if they exist). The conversion is done automatically by Sopa, but it can also be done manually via our CLI: `sopa explorer write <sdata_path> <output_path>`.

For image creation, a Python function is recommended in the Xenium Explorer documentation (https://www.10xgenomics.com/support/software/xenium-explorer/tutorials/xe-image-file-conversion) but is not optimized for large images. We updated it to support Dask[30] arrays, i.e., (the image type used by Sopa). Pyramids of resolutions are generated via the SpatialData library[26]. To decrease memory usage, each (1024 × 1024) image tile is generated using an iterator that only computes the minimally required data from the Dask array at each tile generation. For higher pyramidal levels, where the image size decreases, we allow loading an image into memory if it fits, accelerating conversion.

As transcripts typically cannot be loaded entirely into memory, the Xenium Explorer avoids loading all transcripts. On low-resolution levels, only a subset of transcripts is displayed (subsampled), while zooming in reveals all transcripts from the current field of view. This pyramidal transcript view ensures low memory usage during visualization. The highest-resolution tiles are 250-micron-wide squares. For each pyramid level, the tile width doubles, and only one-fourth of the transcripts from the previous level are retained. The process stops when there is only one remaining tile that is larger than the original slide. Transcript coordinates are stored as separate chunks for each tile and resolution, saved as a Zarr file. This allows loading only the transcripts corresponding to the displayed tiles when zooming in.

Cell boundaries are padded to have the same number of vertices (13). Polygon simplification is applied to polygons with more than 13 vertices using the Shapely library, reducing the number of vertices while preserving shape geometry. A fixed number of vertices enables lighter cell-boundary storage and faster visualization.

Transcript counts (cell-by-gene table) use sparse array storage. One 1D array stores all non-zero transcript counts, another array stores the cell index for each count, and a third array is a pointer indicating the gene index for these counts. Cell categories are similarly saved using indices and corresponding pointers. Once again, the file format employed is a Zarr file.

## Cell-type annotation

**Transcript-based annotation.** Tangram[20] is used for cell-type annotation based on an annotated scRNAseq reference. To make Tangram[20] scalable for large datasets, we adopt a strategy of splitting the data into "bags of cells", with the size determined by the user. This approach ensures that each Tangram iteration operates within manageable memory limits, and we subsequently merge the results to obtain the annotation for the entire dataset. Following this, Leiden[52] clustering can be applied to refine the annotation, associating each Leiden cluster with its most prevalent Tangram cell type. Additionally, we have implemented a multi-level annotation feature based on Tangram to enhance the annotation of minor cell types if needed. The process involves initially annotating global cell populations, followed by running Tangram on specific cell groups (e.g., Myeloid cells) for a more detailed annotation (e.g., pDCs, *TREM2* macrophages, etc.). All that is required is to provide multiple cell-type annotation columns in the reference scRNAseq data, and Sopa will seamlessly execute the multi-level annotation.

**Staining-based annotation.** For non-transcriptomics data, we also provide a fluorescence-based annotation. As each channel intensity is averaged inside each cell, we obtain a matrix $\mathbf{X}$ of shape $(N, P)$, where $N$ is the number of cells, and $P$ the number of stainings/channels. Then, these intensities are preprocessed as in a recent article[53]:

$$\mathbf{X}' = (\mathbf{X}'_j)_{1 \le j \le P}, \text{ with } \mathbf{X}'_j = \text{arcsinh}\left(\frac{\mathbf{X}_j}{5Q(0.2, \mathbf{X}_j)}\right), \quad (1)$$

where $\mathbf{X}'$ is the preprocessed matrix, arcsinh is the inverse hyperbolic sinus function, and $Q(0.2, \mathbf{X}_j)$ is the 20th percentile of $\mathbf{X}_j$. Afterwards, we use a list of stainings corresponding to a population, and each cell is annotated according to the channel whose preprocessed intensity is the highest. If desired, Leiden clustering[52] can be run to have a deeper annotation. Each cluster can be annotated via differential analysis or by showing a heatmap of staining expression per cluster.

## Spatial statistics

All spatial statistics are performed after computing a Delaunay graph based on the spatial location of cells. This is done with Squidpy[22], which is itself based on Scipy[54]. We also prune long edges that cannot correspond to a physical cell-cell interaction (typically, edges longer than 40 microns). In the paragraphs below, $N$ denotes the number of cells.

**Cell category to cell-category statistics.** One relevant spatial statistic is the computation of the mean or minimum distance between two cell categories. This includes the pairwise distance between cell types (e.g., the mean distance between CD8 T cells and tumour cells), as well as the distance between cell types and niches (e.g., the distance between tumour cells and tertiary lymphoid structures). Let $(C_1, ..., C_N)$ represent categories assigned to the $N$ cells (e.g., cell types), and $(C'_1, ..., C'_N)$ represent other categories (such as the niche to which the cell belongs). For instance, if cell "$i$" is a T cell inside the stroma, then $C_i =$ "T cell" and $C'_i =$ "stroma". The sets of unique categories are denoted $G$ and $G'$, respectively; for instance, $G$ can be the set of unique cell types, and $G'$ can be the set of unique niches. Then, $\forall\, g \in G$ and $\forall\, g' \in G'$, we define the mean distance between the category $g$ and $g'$ as follow:

$$D(g, g') = \frac{1}{\text{Card}(\{i \mid C_i = g\})} \sum_{i \mid C_i = g} \min_{j \mid C'_j = g'} d_{ij}, \quad (2)$$

where Card represents the cardinal, and $d_{ij}$ is the hop-distance between cell $i$ and cell $j$. Note that $\min_{j \mid C_j = g'} d_{ij}$ is the distance between cell $i$ and the closest cell of category $g'$, that is how many hops are needed for cell $i$ to "find" the category of interest. In practice, we compute $D(g, g')$ by multi-node graph traversal, starting from all nodes whose category is $g'$. In this way, for each $g' \in G'$, we compute $(\min_{j \mid C_j = g'} d_{ij})_{1 \leq i \leq N}$ in a single graph traversal. All the resulting distances can be stored in a matrix $((D(g, g')))_{g \in G, g' \in G'}$ and shown as a heatmap. Note that this heatmap is asymmetric because of the "minimum" usage in the above distance definition. To prevent confusion while reading the asymmetrical heatmaps, we precise that one row corresponds to the distances from the cell type of the row index to all other cell types. Additionally, we combine the four matrices of distances (cell-type to cell-type, cell-type to niches, niches to cell type, and niches to niches) into an adjacency matrix whose weights are the inverse of the distance. Then, the corresponding network can be plotted using the netgraph[55] library, as in Fig. 4g, providing an interpretable visualization of the tumour microenvironment's structure.

**Niche geometry statistics.** When niches (or spatial domains) are performed with an algorithm such as STAGATE[39], users can decide to extract these niches as geometries to compute some relevant statistics, such as their area, perimeter, or roundness. From now on, for each cell $i$, $1 \leq i \leq N$, $C_i$ denotes the niche to which the cell belongs, and $G$ is the corresponding set of unique niches (i.e., for all cell $i$, $C_i \in G$). First, we prune all the edges $(i, j)$ that are in between niches from the Delaunay graph, i.e., if $C_i \neq C_j$. Then, we extract the connected components of the graph. Because of the way we pruned the edges, each component corresponds to one niche, but one niche can be composed of multiple components (or occurrences). For each component, we search simplices (i.e., triangles from the Delaunay graph) at the component's border, that is, the simplices that have one or two simplex neighbours. From all the border simplices, we extract the corresponding border edges; these edges are then linked to make one or multiple rings (i.e., cyclic lines). If we have only one ring, it is transformed into a polygon, which corresponds to a "full" component. If there are multiple rings, the largest ring is the outer polygon, and the others correspond to "holes" inside the main polygon: this can happen when some components are completely surrounded by another niche. Repeating this process for all components allows the transformation of each niche $g \in G$ into multiple polygons. We can then count how many occurrences (or polygons) each niche is made of, and we can also compute the mean area $A_g$, perimeter $L_g$, and roundness $R_g$ of each niche using Shapely[28]. Note that $R_g = \frac{4\pi A_g}{L_g^2} \in [0,1]$, where higher values correspond to a "circle-like" shape. The density of cells inside a niche is computed as the total number of cells in this niche divided by the total area of the niche. Also, for each niche, we filter out components whose areas are less than 5% of the area of the same niche's largest component, as they usually correspond to low-quality artefacts from the clustering of niches.

**Reporting summary**
Further information on research design is available in the Nature Portfolio Reporting Summary linked to this article.

## Data availability
The MERSCOPE dataset is available online at https://info.vizgen.com/merscope-ffpe-solution. The Xenium dataset is available at https://www.10xgenomics.com/resources/datasets/pancreatic-cancer-with-xenium-human-multi-tissue-and-cancer-panel-1-standard. The Pheno-Cycler dataset is available upon request to Akoya Biosciences, see https://www.akoyabio.com/fusion/data-gallery/. The MACSima dataset is available upon request to Miltenyi Biotec. Source data are provided with this paper.

## Code availability
The code developed in this article is available as an open-source Python package, accessible on Github at https://github.com/gustaveroussy/sopa, or with the Zenodo DOI 11084433[56]. The code used to run the benchmark is available at https://github.com/quentinblampey/sopa_benchmark.

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

## Acknowledgements

This work is supported by Prism—National Precision Medicine Center in Oncology funded by the France 2030 programme, the French National Research Agency (ANR) under grant number ANR-18-IBHU-0002, ARC Foundation and Fondation Gustave Roussy.

## Author contributions

Q.B. implemented the Sopa library, performed the computational benchmark, and wrote the manuscript. K.M. defined the biological aims of the project, performed the biological analysis, and wrote the manuscript. M.G. performed experiments as part of a clinical study involving private data. S.C. implemented capabilities for handling H&E data to the Sopa package. C.A.D. detailed the need for a new method to standardize the analysis of spatial omics data. F.G. supervised K.M. with a particular focus on the biological aspect of the project. P.H.C. supervised Q.B., concentrating on the methodological aspects of the work and manuscript development. Finally, F.A. and P.H.C. acquired the grant to fund the project.

## Competing interests

The authors declare no competing interests.
