## [Peer Review File · Nature Communications]

Reviewers' Comments:

Reviewer #1:

Remarks to the Author:

This paper presents SOPA, a tool for technology invariant analyses of spatial-omics data. The paper is very well written. The authors have analyzed four different datasets and produce insightful analyses. The data availability section clearly points to the datasets. The github repo is well organized. Overall, I think this is a useful tool. Specially the memory benchmark shows that researchers with limited resources will be able to utilize this tool for useful analyses. I recommend publication of this paper with minor modifications.

Here are some minor comments:

1. <https://gustaveroussy.github.io/sopa/pipeline> is not functional
2. Sopa can be installed only with python 3.10. But Python 3.10 is no longer downloadable in Windows from <https://www.python.org/downloads/windows/>. Therefore I was not able to run and check it out. The tool should be supported for higher versions of Python. (The github README does mention "Sopa can be installed via PyPI on all operating systems.", therefore, I'd like to run it on my Windows desktop to check it out.
3. The Xenium dataset location URL is broken. However, it is not hard to navigate to it with a bit of searching.
4. Line 71: than  then
5. Line 243 – But the cell is likely to be in a partial capture in one of the patches as well. Would it impact that patch adversely?
6. Figure 2f: Looks like image writing with sopa is quite expensive. Please discuss this.
7. Lines 158-159: Are the figure references correct? Figure 3c is about MERSCOPE data. Did you mean Figure 3e?
8. Line 195: Figure 4c seems to be incorrectly referred here.
9. Section 3.5 – If one is running in their own desktop/laptop (i.e. no access to high-performance cluster), will they be able to count the transcripts? If not, then this would contradict line 367.

Reviewer #2:

Remarks to the Author:

In this manuscript, the author introduces a tool termed SOPA, which provides a memory-efficient workflow for processing image-based spatial transcriptomics data. SOPA integrates various pre-existing tools, including Cellpose, Baysor, and Tangram, with some statistical analysis (such as conflict resolution and aggregation) to generate a consolidated pipeline for image segmentation, cell type annotation, and derive spatial statistics from the consequent output. While such a contribution does streamline data processing and enhance efficiency and scalability, the research does not reach the distinctive innovative and breakthrough thresholds typically expected for publications in Nature Communications. Therefore, my suggestion aligns with not accepting this paper for publication for the following two reasons:

1. Where novelty is concerned, SOPA appears more as an optimized pipeline for data processing. Although it offers a standardized way to manage datasets across spatial transcriptomic technologies and improves memory efficiency through its integrated workflow, it has not introduced any significantly novel elements or components.
2. Additionally, it is key to highlight that this method does not lead to the discovery of new biological insights that cannot be derived from pre-existing tools. Figure 5 presents some spatial statistics derived from SOPA, however, similar results could be achieved using existing tools.

Below are some suggestions to make the paper more solid:

1. How to choose the 0.2 and 0.8 cutoff? In Figure 2c, the author chose the threshold between good resolutions and bad ones to be 0.2 and 0.8. How are these two numbers determined?
2. In Figure 3, instead of just benchmarking with 10X/Vizgen default, I will also suggest to benchmark SOPA with other tools (such as directly using Baysor or Cellpose without per-patch segmentation) to compare scores.

3. Continuing from Point 2, a comparison of SOPA's results with outcomes from Cellpose/Baysor without the per-patch segmentation could be insightful. One would expect SOPA to enhance memory efficiency while potentially decreasing accuracy. It would be valuable to quantify how much this accuracy is compromised.

Reviewer #3:

Remarks to the Author:

The authors have developed a software package called Sopa, designed to analyze and visualize image-based spatial omics datasets from various technologies, including Xenium and MERSCOPE. Sopa encompasses several data processing tasks such as segmentation, annotation, and geometric/spatial analysis. It is built on the SpatialData framework and can be integrated with scverse. The primary contribution of Sopa lies in its memory-efficient, patch-based segmentation method. The authors also demonstrate the applicability of their approach to multiplexed imaging datasets. Overall, this package holds significant potential for processing datasets from diverse technologies. However, I believe that the analysis and baseline comparisons require further refinement.

Here are my comments:

1. It would be beneficial to include the full dataset names, indicating the tissue/cell line analyzed, in the legend of each figure.

2. Some figure names have been cited incorrectly. For instance, in L158-159, figure 3d and figure 3c have been referenced for the clear delimitation of B cells, yet 3c is not a UMAP plot.

3. The authors claim that patch-based segmentation consumes minimal memory while maintaining similar accuracy to existing segmentation techniques such as Cellpose. However, in fig 3c, significant points indicate poor scores, especially for Cellpose. Furthermore, the authors assume that scores greater than 0.8 indicate good performance without providing supporting visuals or statistics.

4. Expanding on comment #3, the authors should conduct the above experiments on different datasets with varying cell shapes and sizes. The merging of segmented cells is a critical step, and it should be clearly discussed under what scenarios such a step would not introduce significant errors in downstream tasks. Reference to Fu et al. Nat Com 2023, which addresses the handling of cells with varying shapes, and the bioarxiv paper "Comparative analysis of multiplexed in situ gene expression profiling technologies," discussing the impact of incorrect segmentation on cell type annotation, would be beneficial.

5. In figure 3, the authors compare with Vizgen and 10x genomics clusters, which may not be the appropriate baselines. Authors should compare with Cellpose and/or Baysor to demonstrate that there is minimal difference in the Calinski-Harabasz score while using low RAM. If the authors are unable to run the above methods on the full dataset, conducting experiments on a subset of data would be appropriate.

6. Figure 4 and Figure 5 present basic analyses that could be performed using existing toolkits such as sc-verse and squidpy.

In conclusion, while this tool holds promise for preprocessing datasets from different imaging technologies within the SpatialData framework, subsequent analyses such as segmentation may be error-prone and require further investigation.

Reviewer #1 (Remarks to the Author):

This paper presents SOPA, a tool for technology invariant analyses of spatial-omics data. The paper is very well written. The authors have analyzed four different datasets and produce insightful analyses. The data availability section clearly points to the datasets. The github repo is well organized. Overall, I think this is a useful tool. Specially the memory benchmark shows that researchers with limited resources will be able to utilize this tool for useful analyses. I recommend publication of this paper with minor modifications.

We truly appreciate the positive feedback, notably that the work could be useful for the community. We addressed the following comments carefully to improve the manuscript.

Here are some minor comments:

1. <https://gustaveroussy.github.io/sopa/pipeline> is not functional

We thank the reviewer for pointing out the non-functional link. We recently updated our documentation to improve its readability, but we forgot to update this link in our README. This is now corrected and should work as intended.

2. Sopa can be installed only with python 3.10. But Python 3.10 is no longer downloadable in Windows from <https://www.python.org/downloads/windows/>. Therefore I was not able to run and check it out. The tool should be supported for higher versions of Python. (The github README does mention “Sopa can be installed via PyPI on all operating systems.”, therefore, I’d like to run it on my Windows desktop to check it out.

This is right, Python 3.10 cannot be installed on Windows. Therefore, we updated Sopa, which now supports Python 3.9, 3.10, and 3.11, thus covering most of the recent Python versions. We also added some tests to the GitHub Actions of our repository to check for the stability of all these versions, and we performed some tests on Windows users' laptops. You should therefore now be able to install and use Sopa.

3. The Xenium dataset location URL is broken. However, it is not hard to navigate to it with a bit of searching.

It seems that the link was functional but overflowing when rendered on the PDF (we assumed you talked about the URL from the “Data availability” section). We added a line break in order to display the URL entirely on the PDF.

4. Line 71: than  then

It has been corrected.

5. Line 243 – But the cell is likely to be in a partial capture in one of the patches as well. Would it impact that patch adversely?

Indeed, this case can happen. Since the cell is complete in one of the two overlapping patches, it will lead to one “full boundary” in one patch and one “cut boundary” on the partial capture. Yet, the latter “cut boundary” should be entirely included in the “full boundary”. Therefore, when we merge the two cell boundaries, we will have one complete cell boundary with no artefact leftover. These cases correspond to the scores close to 1 in the distribution shown in Figure 2d, whose density is high. We added some visuals to better grasp this mechanism (see Figure 2c). Overall, these figures show that the case you mention does not decrease the segmentation quality because merging the two cells will simply keep the “full” cell. We added some comments on section 2.3 and in the caption of Figure 2 related to this.

6. Figure 2f: Looks like image writing with sopa is quite expensive. Please discuss this.

This is an excellent remark. We investigated in more depth why image writing was time-consuming, and we discovered that it was simply due to a bad choice in the default image chunk size. We changed the default chunk size to 1024, i.e., the chunk size intended by the Xenium Explorer, which gave an x5 speed-up on image writing. Thanks to this correction, the image writing with Sopa is now very fast while still being memory efficient. We updated the manuscript and Figure 2 accordingly.

7. Lines 158-159: Are the figure references correct? Figure 3c is about MERSCOPE data. Did you mean Figure 3e?

Thanks for noticing the inaccurate figure calling, we intended to refer to “supplementary Figure 3c” instead of “Figure 3c”. This has been corrected in the updated manuscript.

8. Line 195: Figure 4c seems to be incorrectly referred here.

Again, we thank the reviewer for pointing out the mistake. We meant “supplementary Figure 4c”. We updated it and checked that this mistake had not been made elsewhere.

9. Section 3.5 – If one is running in their own desktop/laptop (i.e. no access to high-performance cluster), will they be able to count the transcripts? If not, then this would contradict line 367.

Yes, counting the transcripts using a desktop or a laptop is possible. We noticed that section 3.5 was indeed confusing because it did not mention the use of personal machines. We updated the manuscript to explain that our approach is “highly effective on both laptops and high-performance clusters” because “Dask is designed to seamlessly scale these processes without necessitating any code modifications”. Indeed, our benchmark in Figure 2 shows a RAM usage of about 3GB on very large images, while most laptops have between 8GB and 32GB of RAM.

Reviewer #1 (Remarks on code availability):

I reviewed the github repo of the package. The authors also provided a separate repo/link for scripts to reproduce all the results. This seems very organized.

However, Sopa can be installed only with python 3.10 and Python 3.10 is no longer downloadable in Windows from <https://www.python.org/downloads/windows/>. Therefore I was not able to run and check it out. The tool should be supported for higher versions of Python. The github README does mention “Sopa can be installed via PyPI on all operating systems.”, Therefore, I’d like to run it on my Windows desktop to check it out.

We thank the reviewer for the positive comments on the quality and organisation of the repository. As requested, we updated the dependencies of Sopa to make it available on Python versions 3.9, 3.10, and 3.11, which now cover Windows, Linux, and MacOS.

Reviewer #2 (Remarks to the Author):

In this manuscript, the author introduces a tool termed SOPA, which provides a memory-efficient workflow for processing image-based spatial transcriptomics data. SOPA integrates various pre-existing tools, including Cellpose, Baysor, and Tangram, with some statistical analysis (such as conflict resolution and aggregation) to generate a consolidated pipeline for image segmentation, cell type annotation, and derive spatial statistics from the consequent output. While such a contribution does streamline data processing and enhance efficiency and scalability, the research does not reach the distinctive innovative and breakthrough thresholds typically expected for publications in Nature Communications. Therefore, my suggestion aligns with not accepting this paper for publication for the following two reasons:

1. Where novelty is concerned, SOPA appears more as an optimized pipeline for data processing. Although it offers a standardized way to manage datasets across spatial transcriptomic technologies and improves memory efficiency through its integrated workflow, it has not introduced any significantly novel elements or components.

Thank you for your thoughtful review. We value your insights regarding the standardised approach and memory efficiency, and we're grateful for your recognition of these aspects. Regarding your comment that our work lacks novelty, we make it clearer below the novel points introduced by Sopa, and we have also tried to improve the way we presented them in the manuscript.

First, to improve the contribution of Sopa, and to broaden its applicability, in this revised version, we have also incorporated H&E image support and demonstrated the potential for multi-omics analyses in Figure 5, showcasing integration at the single-cell level across (i) transcriptomics, (ii) proteins, and (iii) histology layers. Notably, we show that the two omics layers can provide interpretability to the histology, and pave the way to training multi-modal spatial models.

Also, in addition to its flexibility and optimisation of Sopa, we'd like to emphasize that it truly enables novel post-processing analyses. While we acknowledge that we could have provided more clarity on these novel analyses in the original article, we have since invested significant effort in highlighting the distinctions and advantages of our approach compared to existing tools, as addressed in your subsequent point 2.

Regarding the value of Sopa's pre-processing tools, we believe that they bring a lot of value to the community. While it's true that underlying methods like Cellpose/Baysor were already available, their application to large datasets was either limited or infeasible. Through our research, we've developed sophisticated, memory-efficient tools that empower high-quality analyses across all sizes and types of spatial datasets. To the best of our knowledge, no existing alternative offers such comprehensive technology coverage and optimisation across various tasks, including segmentation, aggregation, annotation, visualisation, and spatial statistics extraction. As a result, Sopa stands to benefit a wide range of users, from smaller labs with limited computational resources to larger organisations conducting high-resolution spatial omics studies. Our GitHub statistics (3000+ downloads, 40+ stars, and 200+ views per day) seem to show that Sopa is quickly gaining popularity in the community and probably fills a gap. Also, Nature Communications has already published several "optimised pipelines" [1][2][3] if their potential impacts on the community could be significant.

We hope these revisions illustrate convincingly that Sopa represents a significant advancement in spatial omics analysis, offering new perspectives and capabilities to the field.

2. Additionally, it is key to highlight that this method does not lead to the discovery of new biological insights that cannot be derived from pre-existing tools. Figure 5 presents some spatial statistics derived from SOPA, however, similar results could be achieved using existing tools.

Even though the analyses of Figure 5 (now called Figure 4) seem similar to those in Squidpy, they are different and provide different insights. We agree that the difference should have been discussed more in the paper, so we added comments in section 2.6 and detailed the differences in supplementary Figure 5. Indeed, while Squidpy's "neighbourhood enrichment" is designed to capture local organisations, the distances in Sopa are more related to the global organisation. In particular, supplementary Figure 5 shows that the neighbourhood enrichment does not capture niche-level information, while Sopa does. Also, the geometries statistics in Figure 4b require converting the niches into geometries (polygons), which, as far as we know, has not been done yet. This conversion to geometries, while apparently simple, requires careful attention to ensure we have clean shapes and handle all edge cases - which is needed to compute these statistics in large-scale studies. We hope that the additional comments and figures will better show these tools' novelty and capabilities.

Below are some suggestions to make the paper more solid:

1. How to choose the 0.2 and 0.8 cutoff? In Figure 2c, the author chose the threshold between good resolutions and bad ones to be 0.2 and 0.8. How are these two numbers determined?

We would like to thank the reviewer for this excellent remark. In the previous version of the manuscript, the 0.2 and 0.8 thresholds were selected arbitrarily. We agree that we need a meaningful manner to determine these numbers. Therefore, we computed the upper threshold using a statistical test whose null hypothesis is that cells are randomly (uniformly) overlapping. Surprisingly, this gives us a 0.7995 cutoff, which is very close to our initial value (more details in section 9.6). The lower cutoff is related to a different phenomenon: we compute it according to the expectation of the overlapping score between two non-touching 3D cells that appear touching when projected on the 2D plane. This gives us a lower cutoff of 0.07, which is lower than 0.2, but the same conclusion remains (as shown in Figure 2d). Moreover, in order to better visualise such cutoffs, we provided additional visuals (see Figure 2c), which show Sopa's capability to resolve conflicts over overlapping patches.

2. In Figure 3, instead of just benchmarking with 10X/Vizgen default, I will also suggest to benchmark SOPA with other tools (such as directly using Baysor or Cellpose without per-patch segmentation) to compare scores.

This is a very interesting suggestion that has also been asked by the reviewer 3. As mentioned in your point 3 below, we compared Sopa to its underlying segmentation tools without the patching process (12 new experiments, supplementary figure 7), see more details in answer to point 3 below. Alongside this, one advantage of Sopa is its flexibility in the sense that any user can plug-in its segmentation of interest. Therefore, it is possible to use any tool (by just passing a function as an argument), and Sopa will include it in the rest of the process, making the segmentation scalable. To make it easy for users, we have added details in our documentation (link: https://gustaveroussy.github.io/sopa/tutorials/advanced_segmentation/#custom-staining-based-segmentation). Therefore, when a better segmentation tool is developed, Sopa will be able to include it quickly. For instance, CellPose v3 was released one month ago with significant novelties and has already been included in Sopa. Overall, Figure 2 and supplementary Figure 7 shows the capacity of Sopa to scale segmentation methods while retaining their segmentation quality.

3. Continuing from Point 2, a comparison of SOPA's results with outcomes from Cellpose/Baysor without the per-patch segmentation could be insightful. One would expect SOPA to enhance memory efficiency while potentially decreasing accuracy. It would be valuable to quantify how much this accuracy is compromised.

Indeed, this is an interesting comparison to perform. Since Baysor and Cellpose cannot be run on large datasets due to their RAM usage, we had to crop the datasets to perform the requested comparison. We performed the crops such that we still have a large image (16000x16000 pixels) while also being able to run Cellpose and Baysor without the patch process. In order to provide an extensive comparison, we compared the patching and the non-patching process for both Cellpose (on the 4 subset datasets) and Baysor (only the 2 spatial transcriptomics datasets). The experiments have been summarised in supplementary Figure 7. Notably, they show a minor change in the segmentation quality, and the UMAPs overlap completely. In order to also provide insights in Figure 3, we extrapolated the Baysor score on the full dataset and added the expected Baysor score if we were able to run it on the whole data (see Figure 3c/f).

[1] Gharibi, H., Ashkarran, A.A., Jafari, M. *et al.* A uniform data processing pipeline enables harmonized nanoparticle protein corona analysis across proteomics core facilities. *Nat Commun* 15, 342 (2024)

[2] Battenberg, K., Kelly, S.T., Ras, R.A. *et al.* A flexible cross-platform single-cell data processing pipeline. *Nat Commun* 13, 6847 (2022)

[3] Lee, L., Yu, H., Jia, B.B. *et al.* SnapFISH: a computational pipeline to identify chromatin loops from multiplexed DNA FISH data. *Nat Commun* 14, 4873 (2023)

Reviewer #3 (Remarks to the Author):

The authors have developed a software package called Sopa, designed to analyze and visualize image-based spatial omics datasets from various technologies, including Xenium and MERSCOPE. Sopa encompasses several data processing tasks such as segmentation, annotation, and geometric/spatial analysis. It is built on the SpatialData framework and can be integrated with scverse. The primary contribution of Sopa lies in its memory-efficient, patch-based segmentation method. The authors also demonstrate the applicability of their approach to multiplexed imaging datasets. Overall, this package holds significant potential for processing datasets from diverse technologies. However, I believe that the analysis and baseline comparisons require further refinement.

We truly appreciate your positive feedback, and we are glad to hear that you found the work useful for the community. Based on your comments, we improved the manuscript and performed some new analyses to match your requirements, as detailed in the answers below.

Here are my comments:

1. It would be beneficial to include the full dataset names, indicating the tissue/cell line analyzed, in the legend of each figure.

We thank the reviewer for the insight, and we agree that providing the full name improves the readability. As requested, this was added to the captions of all figures.

2. Some figure names have been cited incorrectly. For instance, in L158-159, figure 3d and figure 3c have been referenced for the clear delimitation of B cells, yet 3c is not a UMAP plot.

Thank you for noticing this mistake. We wanted to refer to “supplementary Figure 3c” instead of “Figure 3c”. This has been corrected.

3. The authors claim that patch-based segmentation consumes minimal memory while maintaining similar accuracy to existing segmentation techniques such as Cellpose. However, in fig 3c, significant points indicate poor scores, especially for Cellpose. Furthermore, the authors assume that scores greater than 0.8 indicate good performance without providing supporting visuals or statistics.

Actually, both very high scores and very low scores show a good performance. Indeed, a low score (close to 0) happens when two cell boundaries are slightly touching; this can happen because two cells can divide or simply appear as “touching” in images. Oppositely, a very high score (close to 1) is also a good performance because it means that the same cell has been segmented in the same way on two different patches, resulting in a boundary merging without any artefact. Thanks to your comment, we added a new section to compute statistically significant thresholds instead of simply assuming that 0.8 was a “good score”. While the upper threshold remains close, the lower threshold is now smaller but still reasonable, as shown in Figure 2d. This is detailed in section 9.6, and we also added some visuals in Figure 2c to show that Sopa has a strong performance in the conflict resolution of the segmentation.

4. Expanding on comment #3, the authors should conduct the above experiments on different datasets with varying cell shapes and sizes. The merging of segmented cells is a critical step, and it should be clearly discussed under what scenarios such a step would not introduce significant errors in downstream tasks. Reference to Fu et al. Nat Com 2023, which addresses the handling of cells with varying shapes, and the bioarxiv paper "Comparative analysis of multiplexed in situ gene expression profiling technologies," discussing the impact of incorrect segmentation on cell type annotation, would be beneficial.

As detailed in the answer to comment 5 below, we added an extensive study (12 experiments) to compare the impact of the quality of the “boundaries merging” on patches overlap (supplementary Figure 7). This includes the four datasets used in this article, which are composed of four different technologies, over four different tissues, and different image resolutions. Results show a minor impact of the “boundaries merging”, whatever the dataset considered.

Concerning the first article you suggested, we assumed it was about BIDCell (Xiaohang Fu et al., Nat. Commun., 2024). As described in section 2.4, a user can choose to plug BIDCell into Sopa if desired, as well as any other segmentation tool, and Sopa will handle the scalability and the rest of the pipeline. Also, if many Sopa users start using BIDCell, we would officially support it as one of the default segmentations in Sopa alongside Cellpose and Baysor. We added a reference and a mention of BIDCell in section 2.4.

Concerning the bioarxiv paper, we read it, and we found it very interesting. Thank you for the recommendation. We referenced it and added some comments related to it in the discussion of our manuscript.

5. In figure 3, the authors compare with Vizgen and 10x genomics clusters, which may not be the appropriate baselines. Authors should compare with Cellpose and/or Baysor to demonstrate that there is minimal difference in the Calinski-Harabasz score while using low RAM. If the authors are unable to run the above methods on the full dataset, conducting experiments on a subset of data would be appropriate.

This is a very interesting suggestion that has also been asked by the reviewer 2. As you mentioned, we are not able to run the above methods on the full datasets because of the RAM. Therefore, we subset the datasets as you suggested. We run the comparison for both Cellpose (on the 4 subset datasets) and Baysor (only the 2 spatial transcriptomics datasets). These experiments have been summarised in supplementary Figure 7. Notably, they show a minor change in the segmentation quality, and the UMAPs overlap completely. In order to also provide insights in Figure 3, we extrapolated the Baysor score on the full dataset and added an extra bar for the Baysor score if we were able to run it on the whole data (see Figure 3c/f).

6. Figure 4 and Figure 5 present basic analyses that could be performed using existing toolkits such as sc-verse and squidpy.

Thanks for your insights. First, the purpose of Figure 4 (now in Figure 3) was not about how the figures were obtained, as they can indeed be done with Scanpy. Instead, we wanted to demonstrate Sopa's capability to run on multiplex imaging data and, notably, to average channel intensity inside polygons, giving a high cell-level resolution. We agree that having a full figure for this was unnecessary, so we merged the previous Figure 4 with the current Figure 3, as they are consistent in terms of insights.

Regarding previous Figure 5 (now Figure 4), even though these analyses seem similar to those in Squidpy, they are different and provide different insights. We agree that the difference should have been discussed more in the paper, so we added comments in section 2.6 and detailed the differences in supplementary Figure 5. Indeed, while Squidpy's "neighbourhood enrichment" is designed to capture local organisations, the distances in Sopa are more related to the global organisation. In particular, supplementary Figure 5 shows that the neighbourhood enrichment does not capture niche-level information, while Sopa does. Also, the geometries statistics in Figure 4b require converting the niches into geometries (polygons), which, as far as we know, has not been done yet. This conversion to geometries, while apparently simple, requires careful attention to ensure that we have clean shapes and handle all edge cases - which is needed to compute these statistics in large-scale studies. We hope that the additional comments and figures will better show these tools' novelty and capabilities.

In conclusion, while this tool holds promise for preprocessing datasets from different imaging technologies within the SpatialData framework, subsequent analyses such as segmentation may be error-prone and require further investigation.

Reviewer #3 (Remarks on code availability):

Code seems to be accessible and easy to use.

Reviewers' Comments:

Reviewer #1:

Remarks to the Author:

All my feedback have been well-addressed. Therefore, I recommend publication of the manuscript.

Reviewer #2:

Remarks to the Author:

I'd like to thank the authors for addressing the concerns raised by reviewers in their revised manuscript. My primary reservations previously revolved around the novelty factor of the research. The added analysis along with Figure 5 does provide valuable insights and ease these concerns. I agree with the authors and Reviewer 1 on the probable utility of this tool for the scientific community. Overall, I am in support of the publication of this paper. But I propose that the authors address two minor questions.

Minor Queries:

1. I suggest that the authors consider adding the following statistics to the paper: After the implementation of patching with the selected segmentation size, what is the percentage of the cell population affected by a cut at the boundary that results in segmentation conflicts for different datasets? After conflict resolution among this population, what percentage of results differ compared to Sopa without patches (one cell recognized as two separate cell or reverse)?

2. I would consider Sopa without patches as the ground truth for benchmarking since it aligns exactly with Sopa without the necessity to resolve conflicting boundaries. Given this fact, I encountered some surprise when observing the results for one of the metrics (mean DE score in Figure 3c and 3f) displaying a significant poorer performance for Sopa without patches. Could the authors delve deeper into this finding and provide an explanation?

Reviewer #3:

Remarks to the Author:

The authors have effectively addressed two of my primary concerns: 1) establishing a threshold for IOMA, and 2) providing a comparison with Baysor and Cellpose using a subset of data. The inclusion of Supplementary Figure 7, which illustrates that the patch-based method performs comparably well on CHS, is a significant improvement. I would recommend incorporating a subpanel in Figure 2 to highlight this specific finding.

Overall, with these revisions, I find SOPA to be a valuable tool and recommend its acceptance.

Reviewer #1 (Remarks to the Author):

All my feedback have been well-addressed. Therefore, I recommend publication of the manuscript.

Reviewer #1 (Remarks on code availability):

I only scanned through the Github repo. The code is well organized. They have provided separate scripts to reproduce the results from the various experiments.

We truly appreciate the positive feedback on the manuscript and the code quality.

Reviewer #2 (Remarks to the Author):

I'd like to thank the authors for addressing the concerns raised by reviewers in their revised manuscript. My primary reservations previously revolved around the novelty factor of the research. The added analysis along with Figure 5 does provide valuable insights and ease these concerns. I agree with the authors and Reviewer 1 on the probable utility of this tool for the scientific community. Overall, I am in support of the publication of this paper. But I propose that the authors address two minor questions.

Thank you for your positive feedback regarding the revision and the novelty that has been added.

Minor Queries:

1. I suggest that the authors consider adding the following statistics to the paper: After the implementation of patching with the selected segmentation size, what is the percentage of the cell population affected by a cut at the boundary that results in segmentation conflicts for different datasets? After conflict resolution among this population, what percentage of results differ compared to Sopa without patches (one cell recognized as two separate cell or reverse)?

This is indeed an interesting suggestion. This has been performed and added in Supplementary Table 1. As expected, the results show that only a minor percentage of cells differ when patch segmentation is performed.

2. I would consider Sopa without patches as the ground truth for benchmarking since it aligns exactly with Sopa without the necessity to resolve conflicting boundaries. Given this fact, I encountered some surprise when observing the results for one of the metrics

(mean DE score in Figure 3c and 3f) displaying a significant poorer performance for Sopa without patches. Could the authors delve deeper into this finding and provide an explanation?

This is an interesting aspect that we also noticed. Running Baysor on patches simplifies the complexity of each run since it will focus on a subset of cell types, which could increase Baysor specificity. For instance, for a patch that is specific to the stroma, Baysor may have an enhanced resolution compared to a run on the full image (which contains a broader range of cell types). This may explain why it has more power for DE, and it is opening up potential investigations beyond this paper to confirm this explanation. We added this comment in the Supplementary Notes.

Reviewer #3 (Remarks to the Author):

The authors have effectively addressed two of my primary concerns: 1) establishing a threshold for IOMA, and 2) providing a comparison with Baysor and Cellpose using a subset of data. The inclusion of Supplementary Figure 7, which illustrates that the patch-based method performs comparably well on CHS, is a significant improvement. I would recommend incorporating a subpanel in Figure 2 to highlight this specific finding.

Overall, with these revisions, I find SOPA to be a valuable tool and recommend its acceptance.

We are pleased to see that Supplementary Figure 7 has been seen as a significant improvement. As suggested, we incorporated it in Figure 2d.